# A Dysfunctional Legal Framework for Failed Public–Private Partnership Projects: Accounting or Economics?

**Vicente Alcaraz Carrillo de Albornoz *** , **Juan Molina Millán, Antonio Lorenzo Lara Galera**
**and Belén Muñoz-Medina**

Ingeniería Civil: Construcción, Universidad Politécnica de Madrid, Aranguren 3, 28040 Madrid, Spain
* Correspondence: vicente.alcarazc@upm.es

**Abstract:** Public–private partnerships (PPP) are complex long-term arrangements used in public infrastructure, public services or public facilities projects. To ensure that PPPs transfer enough risk to the private sector, European directives first and member states' national legal frameworks later have modified how these projects are treated in the event of early termination. This paper aims to analyze said changes to determine if they well serve the public interest. The analysis is done by mapping the different alternatives a granting authority has when confronted with a PPP early termination, as stated by the law. We illustrate our analysis with a case study: The Móstoles-Navalcarnero railway, a Spanish PPP project that came to early termination during the construction stage. For this project, we determine the options available to the granting authority with the old Spanish legal framework (in force at the time the project was awarded) and with the new Spanish legal framework.

**Keywords:** public-private partnerships; PPP; public law; public accounting; contract early termination; final payment; Móstoles-Navalcarnero railway

## 1. Introduction

A public authority that wishes to implement a new infrastructure, facility or service may do so through ordinary contracts, such as engineering procurement and construction (EPC) contracts. An alternative available to some public authorities is public–private partnerships (PPPs). In a PPP, the private sector is usually responsible for financing, constructing, operating and maintaining the underlying asset of the project (such as a road or a railway for instance), bearing an important share of the associated risks. In exchange for carrying out these activities and retaining an important part of the risks associated with the project, the private partner is entitled to economic rights that are usually related to the future cash flows of the project (Grout 1997; Yescombe 2007). A very well-known type of PPP is for instance toll-highway concessions, in which users have to pay to access and use the infrastructure.

In Europe, especially during recent decades, PPP arrangements have been increasingly popular. The reason that justifies this trend is the ultimate goal of achieving a higher efficiency of public funds, although in many cases, PPPs have been an instrument to avoid budget constraints preventing infrastructure programs from being accomplished (Vickers and Yarrow 1991; Ortega et al. 2016).

Although PPPs are usually long-term arrangements intended to last several decades, sometimes they come to early termination. The project may stop during the construction or operation stage. The causes leading to early termination may be attributable to the private partner, the granting authority or both (EPEC 2013). When a PPP project comes to an early termination during the construction stage, this is usually due to construction and expropriation cost overruns (Albalate and Bel-Piñana 2016). When a toll motorway PPP project comes to an early termination during the operation and maintenance stage, this is usually due to traffic below expectations (Flyvbjerg et al. 2005; Bain 2009; Vasallo and Ortega 2011).

Once a PPP contract comes to early termination, the private partner is usually entitled to a final payment from the granting authority to prevent the unfair enrichment of the public sector since the private partner loses all the rights that it had (and in particular economic rights) over the underlying asset. This doctrine is one of the fundamental pillars of public law in many countries (see for instance Birks 2002).

The mechanism describing, for a particular PPP contract, the quantification and payment of this final settlement in the event of early termination is provided by the PPP contract and the law. In this regard, the European Union issued Directive 2014/23/EU, which among other things aims to ensure that there is an effective transfer of operating risk on concessions. This directive has later been incorporated into member states' legislation, usually modifying previous provisions regarding the early termination of PPP contracts.

This paper analyses the changes introduced in public procurement Laws, focusing on the Spanish case. The paper begins by identifying the research gap, then states the research question and the purpose of the paper, continues describing the research method and finally, applies the research method to a case study. Thus, the Section 1 of the paper describes the current state of the art. The Section 2 analyses the changes from a legal and economic perspective, to identify potential discrepancies. The Section 3 presents a real case study, the Móstoles-Navalcarnero railway project, to illustrate the results from the Section 2. The Section 4 discusses the findings of the analysis.

## 2. Literature Review

PPPs have attracted a lot of attention from academicians in the last decade. There currently exists an important body of scientific knowledge about PPPs that can be categorized into four groups: the PPP concept, the risk-sharing amongst PPP participants, the drivers of PPP adoption and the performance of PPPs (Wang et al. 2018). Some of the research applies techniques widely used in other sectors to the particular case of infrastructure and PPPs—such as real options (Lara et al. 2016).

Failed PPPs during the operation and maintenance stage are well documented, with a significant number of papers and studies that analyze the factors behind the gap between the expected outcomes and the actual outcomes (Vassallo et al. 2012; Albalate and Bel-Piñana 2016; Engel et al. 2015). Most recently, a paper analyzing to what extent these PPP projects are a failure and what to do in this situation was written (Alcaraz et al. 2019).

Many academic articles are describing current practices and propose new methods to determine the final payment that the granting authority should pay to the exiting concessionaire in case of PPP contract early termination (see for instance Caselli et al. 2009; Xiong et al. 2016).

There is also abundant literature covering the changes to Spain's Public Contracts Act from 2017, highlighting its strengths and weaknesses (Gimeno Feliu 2017; Fernández Valverde 2017; Lora 2017). The strengths are usually related to a higher risk transfer to the concessionaire, leading in turn to the projects not being included in the government's accounts. The weaknesses have to do with difficult-to-apply processes in practice, with the denaturalization of the different types of contracts to comply with national accounting standards and with a margin for improvement concerning efficiency and transparency in public procurement processes.

There isn't however to the best of our knowledge a study that considers the changes introduced by the European directives in the Public Contracts Acts and the economic logic of early termination of PPP contracts.

## 3. Analysis of the Early Termination Event in PPP Contracts

In this context, it is quite relevant to analyze the changes introduced by the European directives in the Public Contracts Acts, and how well they serve the public interest. This section performs modelling of the PPP contract early termination process, first from Spain's legal perspective and then from an economic perspective. The two models are then compared to identify potential discrepancies.

Although there is an impressive array of tools and literature concerning private business modelling (see for instance Aguilar-Savén 2004), not much attention seems to have been paid to public process modeling, except for E-government (see for instance Palkovits and Wimmer 2003). In the end, the technique used in this paper is a flowchart, which can be defined as a diagram representing a set of activities involved in a process.

### 3.1. Legal Perspective

In the context of a European Union member state, the course of action regarding the early termination of a PPP contract depends on whether the contract comes to an early termination due to the public authority or due to the concessionaire. Figure 1 provides an interpretation of the consequences derived from the early termination of a PPP contract according to Spain's Public Contracts Act from 2017 (LCSP). This law incorporates several novelties in comparison with the precedent Public Contracts Act from 2011 (TRLCSP), in order among other things to comply with European Directive 2014/23/EU.

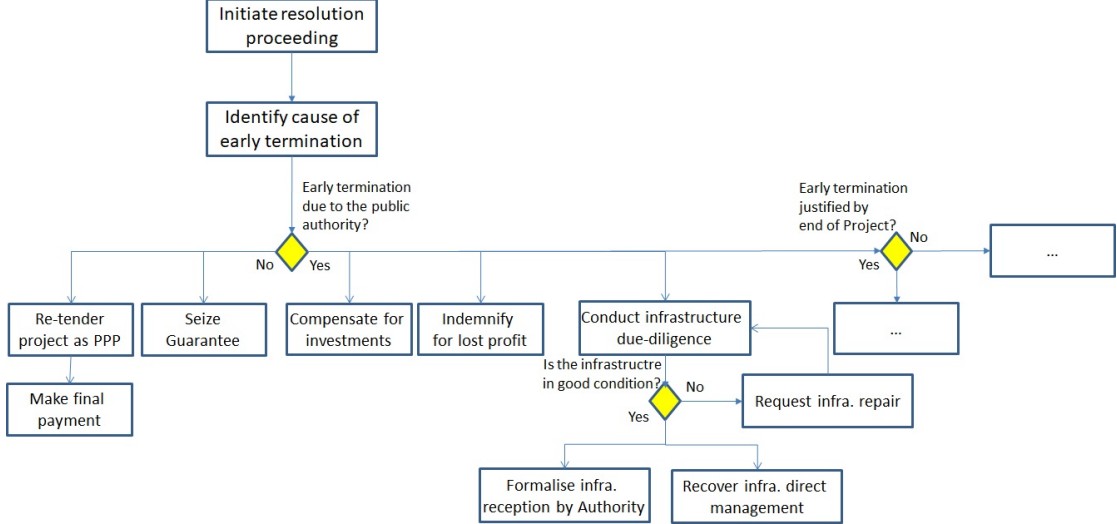

**Figure 1.** Spanish Legal framework for PPP projects that come to early termination. Source: Self-elaboration.

The LCSP points, although not explicitly, to the initiation of a formal resolution proceeding that results in the termination of the PPP contract and that must be concluded in any case before the PPP contract is re-awarded to a different concessionaire (Article 281.1).

It is also clear from the LCSP that this resolution proceeding must explicitly determine the cause that led to the early termination of the PPP contract. This is a critical aspect of the early termination process because the final payment to be made by the granting authority to the exiting concessionaire and the future of the PPP project depend on whether the early termination was due to a cause attributable to the concessionaire or to the public sector (Article 280).

If the PPP contract comes to early termination for a cause attributable to the public sector, then the final payment made by the granting authority (Article 280.1 and 280.3) does not differ much from the payment made under the precedent version of the law (TRLCSP, Article 271). In this scenario, the PPP contract comes to early termination, but also the PPP project seems to be terminated (Article 283), implying that the granting authority recovers the direct management and all the rights related to the underlying infrastructure, following a procedure designed to ensure that the assets are in good shape (Article 243). Since the law does not specify anything else, it can be presumed that the granting authority has then the freedom to do as it considers appropriate with the project: stop it, manage it directly, or manage it indirectly through a new PPP.

If the PPP project comes to early termination for a cause attributable to the concessionaire, then the PPP contract comes to an early termination but the PPP project is not terminated. Instead, the granting authority is forced to re-tender the PPP project (Article 281.1). From a legal perspective and to all effects, the new concessionaire will step into the position of the exiting concessionaire, in particular in issues as critical as the length of the contract, or the risk transferred from the public sector to the concessionaire (Article 281.3). The new PPP contract is awarded through an auction (281.1), and the proceedings from the auction constitute the indemnity to be paid to the exiting concessionaire (Article 280.1). The process seems to resemble what is contemplated in the Spanish Bankruptcy Law for private ventures that do not have a PPP contract as their main source of business and that undergo severe financial difficulties.

### 3.2. Economic Perspective

In this section, we analyze the different issues faced by a granting authority that has to deal with the early termination of a PPP project, assuming there are no legal constraints. These issues include at least:

- Determining whether to continue with the project.
- If the project is to continue, establish the most efficient arrangement.
- Carrying out a final payment to the exiting concessionaire.

From an economic perspective, and for self-financing projects, the most logical course of action is the one depicted in Figure 2. Projects that are not self-financing would need additional studies to determine their affordability for the public sector.

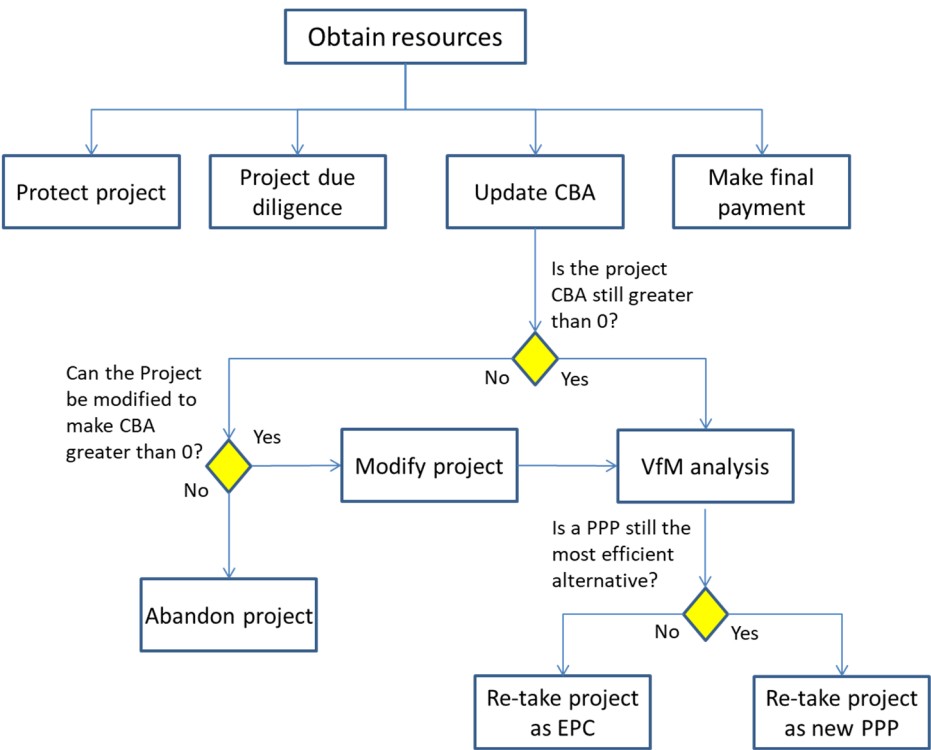

**Figure 2.** A proposed framework for PPP projects that come to early termination—from an economic perspective. Source: Self-elaboration.

The first step for a granting authority in such a situation is to obtain the necessary human and economic resources needed: firstly, to protect the project (construction site if the project was in the construction stage, or the infrastructure or public service if the project was operational) and prevent its deterioration; secondly, to conduct a project due diligence;

thirdly, to conduct the necessary studies to determine what to do with the project; and finally, to make the final payment to the exiting concessionaire.

The protection of the project is essential because the studies that are to be carried out by the granting authority, and even the effective termination of the PPP contract, may take several months. Even if the project is eventually retaken as a new PPP arrangement, the preparation of the new contract and the tendering process will take additional time. If for instance the PPP project was in the construction stage and is left unprotected, the construction site may be a potential source of accidents and will undoubtedly deteriorate (see for instance Doraisamy et al. 2015). It is thus a priority to protect the site until a new contractor arrives to either demolish or retake the management of the infrastructure.

Site due diligence should be conducted as well. This will allow appraising the quantity and the quality of the works effectively executed by the exiting concessionaire in case the early termination of the PPP contract takes place during the construction stage, or the condition of the infrastructure and other related assets if the project has reached the operational stage. Flyvbjerg (2013) suggests how this may be conducted in the case of a PPP project. This information is essential to value the investment that has been done by the concessionaire, which may be (or not) crucial to determine the final payment to be made to the exiting concessionaire. It is also crucial to determine the state of the assets that will be handled back to the granting authority and to quantify the required capital expenditure to eventually complete the construction if the project was in its construction stage or to ensure that the infrastructure is in good operating conditions if the project was on its operational stage.

In case the early termination was not caused by the public sector, the granting authority should probably update some of the studies that were carried out during the pre-contractual stage of the project, as to decide what to do next. To begin with, the cost-benefit analysis should be updated to determine if the project should be continued or abandoned. The initial circumstances that motivated the project may have changed. The sunk costs should be taken out of the study. There is ample practitioner and academic literature on cost–benefit analysis (Layard and Glaister 2003; Priemus et al. 2008).

If the cost–benefit analysis shows a positive balance for the project, then the public authority still needs to determine if the project should be continued as a PPP or as an ordinary public works contract (usually referred to as engineering, procurement and construction or EPC contracts) or services contract. The value for money analysis (VfM) should be updated, taking into consideration the investment already undertaken and the remaining risks. Although VfM is not exempt from controversy (see for instance Grimsey and Lewis 2005; Leighland 2006), and even if the EPC option is not available, carrying out a VfM might prove useful to determine the most appropriate risk allocation for the project.

Finally, an important advantage of delivering the project as a new PPP is that the proceeds obtained in the process could be used to offset the final payment that must be fulfilled to the exiting concessionaire. This can prove critical if the public sector did not set some resources aside for failed PPP projects, as is often the case. There is ample academic and practitioner literature on the treatment of accounting liabilities and best practices (Checherita and Gifford 2007; Posner et al. 2009; Walker 2009).

### 3.3. Results

In this section, a comparison between the legal and the economic process is established to assess any potential areas of improvement when dealing with the early termination of PPP contracts.

The main difference between the two approaches, as depicted in Figures 1 and 2, has to do with the options left to the granting authority in case of early termination due to causes attributable to the concessionaire.

The economic perspective suggests that an evaluation is carried out to determine if it is still desirable to continue with the project or if it should be stopped. In case it is desirable

to continue with the project, further studies should be completed to determine the most efficient arrangement for the project—direct management or a new PPP for instance.

The law determines that there is only one possible course of action: continue with the PPP project.

This can potentially lead to situations in which the course of action for a project that was delivered as a PPP and comes to early termination is not the most efficient one—at least from an economic perspective.

## 4. Case Study Analysis: Móstoles-Navalcarnero Railway

To better illustrate the differences between the economic and the legal approach to the early termination of a PPP Contract, a Spanish case study is provided in this section: the Móstoles-Navalcarnero railway.

The Spanish case represents an interesting experience when analyzing PPPs in transport infrastructure because of the great number and the great range of outcomes that resulted from collaborations between the public and the private sectors. When it comes to railways, the investment done through PPP schemes from 2003 until 2017 amounts to EUR 2.3 billion (SEOPAN 2018).

Although the case study presented in this paper was awarded before Spain's Public Contracts Act from 2017 was in force, the analysis will consider the project under this new framework.

### 4.1. The Móstoles-Navalcarnero Railway Project

The Regional Government of Madrid prepared an infrastructure project that would connect through a railway line the municipalities of Móstoles (205,000 inhabitants) and Navalcanero (26,300 inhabitants), as shown in Figure 3. The railway line had 14.4 km in length (of which around half would run underground), 6 train stations (of which four were situated underground) and an estimated capital expenditure of EUR 362 million (including VAT). According to the feasibility studies carried out by the granting authority, the line would have 14,000 trips per day. Many of these users would come from future urban developments in Navalcarnero, expected to bring around 70,000 new residents to this municipality.

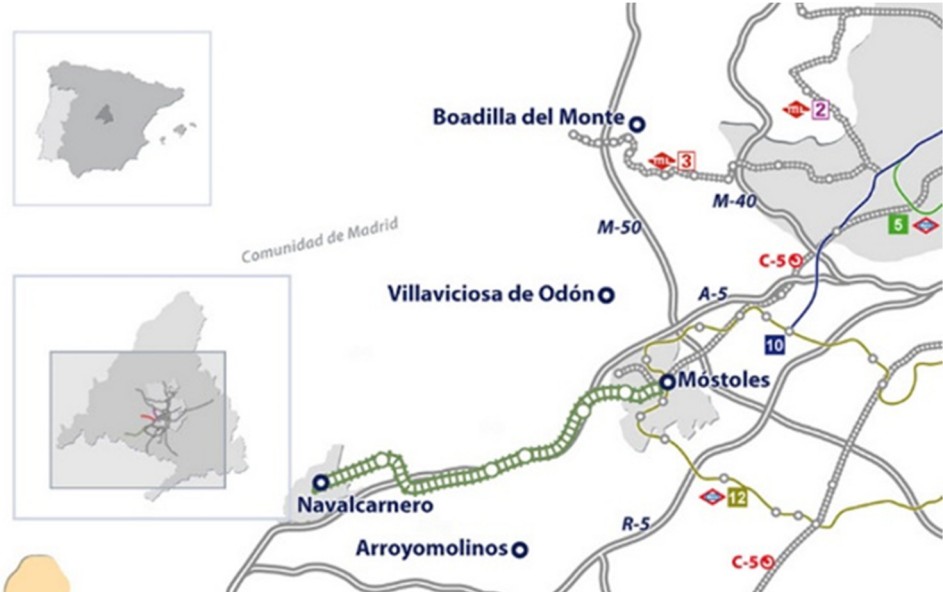

**Figure 3.** Planned itinerary of the Móstoles-Navalcarnero railway. Source: Vialibre La Revista del Ferrocarril.

### 4.2. The PPP Contract

On May 2007 the railway project was tendered as a PPP for the construction, operation and maintenance of the railway line.

Demand risk was to be transferred to the private sector: the Concessionaire was to perceive retribution from the Granting Authority depending on the number of users. This retribution per passenger was an important tendering parameter, with a maximum established at EUR 4.45 per passenger.

The Contract also established that at some point during the construction stage the Concessionaire would receive EUR 50 million from Mostoles' Municipal Government to pay for the burial of the tracks as they passed through the Municipality.

The duration of the contract was 20 years.

### 4.3. The PPP Contract Award and the First Difficulties Encountered

On 23 October 2007, the PPP project was awarded to OHL, a Spanish construction company. On 2 January 2008, the PPP Contract was signed between Madrid's regional government and CEMONASA, a Special Purpose Vehicle (SPV) belonging to OHL and created Ad Hoc to develop the PPP project.

The PPP contract was awarded before the financial close of the project, a usual practice in Spain and many other countries. However, the concessionaire did not manage to achieve the financial close of the project after the contract award. At the time (end of 2007 and early 2008) Spain and many other countries were suffering a severe economic recession.

Nevertheless, CEMONASA had some financial resources—the equity provided by its shareholders—and nothing on the PPP contract established the financial close of the project as a precedent condition to the beginning of the construction stage. Thus, the public works began in February 2008 and were supposed to end in July 2010. This would provide the Concessionaire with some extra time to achieve the financial close of the project.

Unfortunately, CEMONASA ran out of funds before the financial close of the project could be reached. At the beginning of 2010, with approximately 30% of the public works completed, the Concessionaire stopped the construction.

Furthermore, probably also due to the economic crisis, the urban developments in Navalcarnero were considerably fewer than expected. Only 10,000 new residents moved to this municipality, that is, 15% of the forecast made at the beginning of the project. This would undoubtedly have an impact on the number of users of the railway.

### 4.4. The Conflict between CEMONASA and Madrid's Regional Government

In April 2010, the granting authority initiated a sanctioning procedure to CEMONASA for stopping the construction of the railway line, which never concluded.

In July 2010, the concessionaire requested for the first time the early termination of the PPP contract, on three basis: that the Municipality of Móstoles never disbursed the EUR 50 million it was supposed to pay to the SPV to help finance the underground construction of part of the railway line; that there had been modifications to the original project that increased the initial budget in more than 20%; and that it was no longer economically feasible to operate the line considering the foreseeable drop in potential users due to the economic crisis. Nevertheless, in October 2010, the concessionaire withdrew this request.

In December 2010, the granting authority approved a new public works schedule, that envisaged the finalization of the railway line on 31 January 2013. However, CEMONASA never resumed the construction of the project.

In March and July 2014, the granting authority formally requested the concessionaire to resume the construction works, to no avail.

On 12 June 2015, the concessionaire officially requested the early termination of the PPP contract, due to unforeseeable circumstances and a breach of the contract by the granting authority. The concessionaire requested a final payment of EUR 285.44 million from the granting authority: EUR 238.9 million in the concept of the investments made, and EUR 46 million in the concept of loss of earnings. The granting authority denied the

request from the concessionaire and urged the SPV to resume the public works (that had been paralyzed for 5 years) and finish them according to the project.

In December 2015 the concessionaire took the contract to court, requesting the early termination of the PPP contract, and final payment from the granting authority amounting to EUR 369.5 million in the concept of investments made plus damages.

On 11 February 2016, the granting authority gave 16 months to the SPV to complete the construction of the railway and imposed a penalty of EUR 34.08 million for not meeting the deadlines. It also tried to execute the EUR 18 million guarantee that was established by CEMONASA at the time the contract was signed—without success: the concessionaire filed an administrative appeal against the fine, along with a request for its precautionary suspension.

### 4.5. The Early Termination of the PPP Contract

In May 2016 Madrid's Superior Court of Justice denied the precautionary suspension of the fine, and Madrid's regional government seized the 18 million Euro guarantee that CEMONASA had established at the signature of the PPP contract. Immediately afterwards the Concessionaire filed for bankruptcy. In December 2016 the concessionaire started dismantling the tunnelling equipment.

In March 2017 CEMONASA entered into liquidation. In July 2017 it sued the Granting Authority requesting a final payment amounting to EUR 371 million, for the early termination of the PPP contract due to causes attributable to the Granting Authority: the existence of a tacit withdrawal from the Granting Authority concerning the execution of the Contract, and modifications to the original project exceeding 20% of the initial budget, amongst others. On March 2018 Madrid's Supreme Court of Justice rejected the early termination of the PPP contract requested by the Concessionaire because the reasons provided by the SPV that would justify the early termination of the contract were not valid.

On October 2018 the Granting Authority requested that a penalty of EUR 356 million be paid by the Concessionaire, in the concept of damages caused by the interruption of the programmed public works. OHL took to court the issue. In July 2019 Madrid's Supreme Court ruled against the Granting Authority since the PPP Contract was extinguished when CEMONASA filed for bankruptcy in 2016. The ruling also declared void the Granting Authority's resolution in which the PPP Contract was declared terminated for a guilty breach of contract by the Concessionaire.

### 4.6. Current Situation of the Project and Future Course of Action

The hand back of the construction site took place between 2017 and 2018. On 12 July 2018, Madrid's Regional Government spent close to EUR 1 million (VAT included) to protect and maintain the construction site. As a result, the granting authority publicly recognized a liability of around EUR 120 million in the concept of final payment for early termination of the PPP contract. This quantity was deemed too low by the shareholders of the SPV.

In March 2022, and as a result of a court order, the Granting Authority had to pay EUR 162.5 million to the shareholders of the SPV, as a result of the project's early termination.

At this point, the Granting Authority should probably determine if it wishes to continue and finish the construction of the railway project, or if it should stop and demolish what has been done. This can be done because the project was awarded under the Spanish Public Contracts Act of 2000 (TRLCAP), and there were no potential conflicts from the legal and economic perspective of PPP projects coming to early termination. It is noteworthy that, had the project been tendered after the LCSP from 2017 was in force, the Granting Authority would have been obliged to continue with the project, as a PPP.

### 4.7. Simplified Cost-Benefit Analysis for the Railway Line

A cost-benefit analysis (CBA) comparing both scenarios—a scenario in which the project is abandoned and what has been constructed is demolished and a scenario in which

the construction is completed and the railway becomes operational—is probably the best tool available to determine the most efficient course of action for the Public Sector.

A simplified CBA has been carried out, in which the investment done so far is considered a sunken cost and the new demand is taken into consideration. The main hypothesis and data used in this CBA can be found in Table 1.

**Table 1.** Main inputs used in the simplified CBA analysis of the Móstoles-Navalcarnero railway. Source: Authors.

| Input | Value | Source |
|---|---|---|
| Passengers per day | 4000 | News about the project |
| Origin of passengers | | |
| Public transport—bus | 90% | Estimate from authors |
| Private transport—car | 10% | Estimate from authors |
| Capital Expenditure (EUR Million) | | |
| To complete construction | 264.31 | News about the project |
| To demolish public works | 20.00 | Estimate from authors |
| Commercial speed | | |
| Public transport—bus (km/h) | 30.00 | Estimate from Consorcio Regional de Transportes de Madrid |
| Private transport—car (km/h) | 45.00 | Estimate from Consorcio Regional de Transportes de Madrid |
| Public transport—train (km/h) | 45.00 | Estimate from Consorcio Regional de Transportes de Madrid |
| Value of time (Euro/hour) | 10.42 | Metro de Madrid |
| Operating cost | | |
| Public transport—bus (€/1000 km) | 1086.00 | World Bank—HDM VOC |
| Private transport—car (€/1000 km) | 195.24 | World Bank—HDM VOC |
| Public transport—train (Million €/year) | 17.45 | Concessionaire |
| Period of analysis (years) | 50 | European Commission (2014) |
| Social discount rate | 5.0% | European Commission (2014) |

The results of the simplified CBA show a difference between scenario A (abandon project) and scenario B (finish construction of the project) of EUR −377 million, as shown in Figure 4. This would suggest that abandoning the project is the best option for the granting authority since the benefits obtained by the new railway (time savings, reduced pollution and emission of greenhouse gases) do not make up for the costs needed to finalize and operate the project.

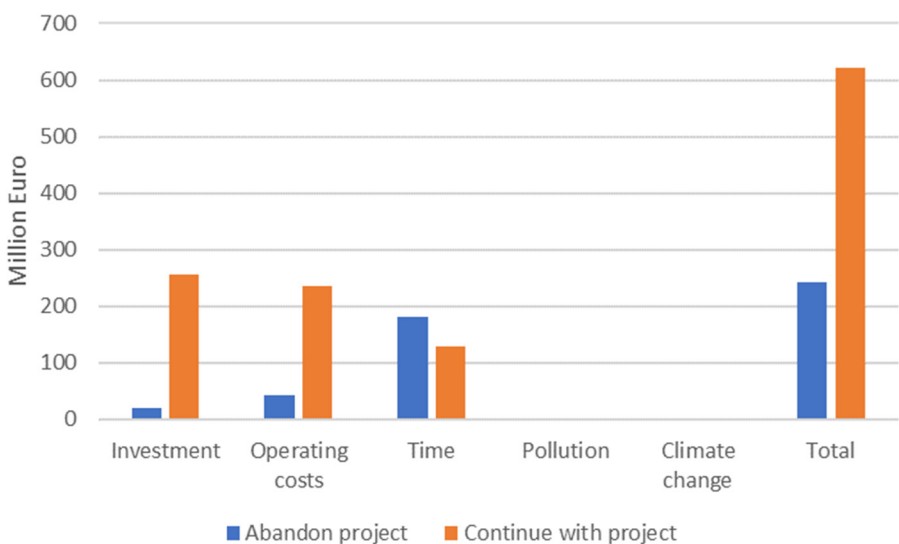

**Figure 4.** Results of the simplified CBA. Source: Authors.

In other words, a reduced number of users would not justify the project. According to the data used, it would be necessary to have at least 20,500 passengers per day (5 times

as many users as it is reasonable to assume that would be using the train) to reach the indifference point between scenarios.

This CBA has many limitations for obvious reasons. The results, however, are so unfavorable to the scenario in which the railway is completed that it is very unlikely that there would be a significant change in the outcome of the study even using a more refined model.

The point of this section is to illustrate that the economic logic would suggest the abandonment of the project, something that would not have been possible for the granting authority to do had the project been tendered under the current law (LCSP).

## 5. Discussion and Conclusions

It is a common practice in PPP arrangements to have a contractual clause establishing a final payment from the granting authority to the concessionaire in the event of early termination. This contractual provision seems to serve two purposes: it prevents the unjust enrichment of the granting authority in case of early termination, and it effectively decreases the risk level supported by the private sector, thus decreasing the cost of capital.

Many countries within the European Union, including Spain, have made changes to their PPP laws introducing a new way to determine this early termination final payment to the concessionaire. Should this early termination happen for a cause attributable to the concessionaire, the final payment is the market value of the PPP contract. This, in turn, forces the continuation of the PPP project with a new concessionaire.

The logic behind this change seems to emanate from the changes introduced by the European System of Accounts, which declares that a PPP should be included in the government's accounts in the existence of grantor financing or granting guarantees, or of advantageous termination clauses notably on termination events at the initiative of the operator" (Eurostat 2013). Furthermore, Eurostat considers that if the granting authority has the option to recover the assets at a predetermined price—and in particular at a price that is not related to its market value—then the assets (and associated liabilities) should be included in the government's accounts (Macho Pérez and Marco Peñas 2014).

Another obvious advantage of automatically re-tendering failed PPP contracts (when early termination is attributable to the concessionaire), in addition to ensuring that they are not included in government accounts, is finding part of the money that will be paid to the exiting concessionaire.

These changes, however, have a serious drawback as well: they may break with the economic logic of the project itself. What if it is not desirable to continue with the project? What if, even if it is desirable to continue with the project, a PPP arrangement is not the most efficient alternative? Although these situations may seem far-fetched, the truth is that they may happen more often than we think, as shown by the Móstoles-Navalcarnero railway project.

In this context, it should probably be highly beneficial to find a way to amend the law to achieve the best of both worlds: having PPPs that do not compute in public accounts because they transfer enough risk to the concessionaire while preserving the flexibility of the granting authority regarding what to do with the project in case of early termination.

It is not the goal of this paper to provide a solution to this problem but merely to point it out so academics and practitioners can start working on it. Some research lines concerning this issue, however, could be the following: gaining a better understanding of the value drivers of PPP projects, so there is an objective way to obtain a proxy to the "market value" of a failed PPP project without the need to re-tender the project itself; challenging the notion that there may be an unjust enrichment of the granting authority if the final payment to the Concessionaire is below the market value of the assets recovered upon Contract termination.

**Author Contributions:** Conceptualization, V.A.C.d.A.; Methodology, V.A.C.d.A. and A.L.L.G.; Formal analysis, J.M.M.; Investigation, V.A.C.d.A., J.M.M. and B.M.-M.; Data curation, J.M.M.; Writing—original draft, V.A.C.d.A.; Writing—review & editing, B.M.-M.; Supervision, V.A.C.d.A. and A.L.L.G. All authors have read and agreed to the published version of the manuscript.

**Funding:** The APC was funded by Universidad Politécnica de Madrid and Fundación Agustín de Betancourt.

**Institutional Review Board Statement:** Not applicable.

**Informed Consent Statement:** Not applicable.

**Data Availability Statement:** The data presented in this study are contained in the article or extracted from the sources cited in the References section.

**Conflicts of Interest:** The authors declare no conflict of interest.

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
