# Peer review of "A Dysfunctional Legal Framework for Failed Public–Private Partnership Projects: Accounting or Economics?"

_socsci, doi:10.3390/socsci11120554_

Round 1

Reviewer 1 Report

The paper is interesting, but a broad introduction of public transportation related PPP investment projects is necessary to make the entire paper understandable to even the professional scientist readers. The question the authors try to analyze are very important on the entire EU level, and also should be made accessible by writing articles such s this to inform the public about these methodologies. The case study is too short.

Author Response

We would like to begin by thanking the reviewer for the time taken and the constructive comments. We are often involved in the process of academic paper review ourselves, and we know it’s not always easy.

We have conducted an extensive English language review. All the changes to the paper have been marked.

Our answers to the reviewer’s comments appear in red.

Reviewer 1

The paper is interesting, but a broad introduction of public transportation related PPP investment projects is necessary to make the entire paper understandable to even the professional scientist readers.

While we feel we provide a good summary of what PPPs are about in the “Introduction” section, it is true that maybe some readers could not be entirely sure about what we are discussing. We have introduced in the first paragraph an example of PPP to help with this issue:

“…project (Grout, 1997; Yescombe 2007). A very well-known type of PPP is for instance toll-highway concessions, in which users have to pay to access and use the infrastructure.

The question the authors try to analyze are very important on the entire EU level, and also should be made accessible by writing articles such as this to inform the public about these methodologies. The case study is too short.

We thank the reviewer for this comment, and we understand that the case study has been interesting and helpful to support the point we are trying to make with a real example.

The case study goes from page 6 (line 207) to page 10 (line 373) on a 12-page paper (416 lines). Almost half of the paper is devoted to the case study. Although we could indeed expand the case study, we believe that we cover the main points and that with the current length the paper is well-balanced between the theoretical and the practical part. We prefer not to expand the case study.

Reviewer 2 Report

1. In the abstract, I propose to emphasise the aim of the work, the research question posed or the hypothesis. The abstract must focus on objectives, mention how they were achieved, and emphasize the results obtained.

2. You have a very good work, deserving publication. Nonetheless, some amendments in the format are due to admit publication. The main problem is the epistemological structure (why the article was conceived and how the study was developed). I suggest the following structure of objectives: (i) research gap; (ii) research question; (iii) purpose of the article; (iv) intermediate objectives; (v) assumptions or hypo; and (vi) research method. This structure must appear in the introduction.

3. The research gap must be created by a systematic literature review that provides 'holes' in the state of knowledge on the topic. I believe that a full review should not be done, but an analysis of about 5-8 studies on the topic under discussion. You can find some examples, which will show the relevance of the issue, as it is indeed a topic of current, relevant research. At the end of the justification you should write something like: According to what we were able to find, there are no studies referring and reporting on ... With this you have therefore proven that the issue is relevant, and you have also proven that your study does indeed fill a research gap. 

4. In conclusion, I propose

-evaluate the critical research, show its limitations and weaknesses,

- highlight the new knowledge and the lessons learned from it,

- describe the importance of the research and how it affects the wider field, show how the information obtained can be further used

Author Response

We would like to begin by thanking the reviewer for the time taken and the constructive comments. We are often involved in the process of academic paper review ourselves, and we know it’s not always easy.

We have conducted an extensive English language review. All the changes to the paper have been marked.

Our answers to the reviewer’s comments appear in red.

Reviewer 2

  1. In the abstract, I propose to emphasise the aim of the work, the research question posed or the hypothesis. The abstract must focus on objectives, mention how they were achieved, and emphasize the results obtained.

Thanks for this comment. We have re-read the abstract carefully and made some changes to our English to improve readability.

We have introduced a methodological sentence in the abstract to illustrate how we proceed to address the research question:

This paper aims to analyze said changes to determine if they serve well the public interest. The analysis is done by mapping the different alternatives a Granting Authority has when con-fronted with a PPP early termination, as stated by the law.

  1. You have a very good work, deserving publication. Nonetheless, some amendments in the format are due to admit publication. The main problem is the epistemological structure (why the article was conceived and how the study was developed). I suggest the following structure of objectives: (i)research gap; (ii) research question; (iii) purpose of the article;(iv) intermediate objectives; (v) assumptions or hypo; and (vi)research method. This structure must appear in the introduction.

Thanks for this comment. The structure we have used for our paper is the following:

  1. What are PPPs and the existence of a new Directive concerning PPP early termination (Introduction)
  2. Research gap (State of the Art) –lines 68 to 70.
  3. Research question and purpose of the article: We have introduced the following phrase at the beginning of Section 3 (Analysis of the Early Termination Event in PPP Contracts) to make more explicit our research question and the paper’s purpose. It now reads as follows:

In this context, it is quite relevant to analyze the changes introduced by the Euro-pean Directives in Public Contracts Acts, and how well they serve the public interest.

  1. Intermediate objectives and assumptions or hypotheses: We do not feel that this applies to this paper.
  2. Research method: This is also described in section 3, in lines 74-77.

Thus, the structure we use in our paper is the one proposed by the reviewer. Following this suggestion, we have modified the final paragraph of the Introduction section, which now reads as follows:

The paper begins by identifying the research gap, it then states the research question and the purpose of the paper, it continues describing the re-search method and finally it applied the research method to a case study.

  1. The research gap must be created by a systematic literature review that provides 'holes' in the state of knowledge on the topic. I believe that a full review should not be done, but an analysis of about 5-8 studies on the topic under discussion. You can find some examples, which will show the relevance of the issue, as it is indeed a topic of current, relevant research. At the end of the justification you should write something like: According to what we were able to find, there are no studies referring and reporting on ... With this you have therefore proven that the issue is relevant, and you have also proven that your study does indeed fill a research gap.

We agree that this is how the research gap should be illustrated. We also believe that this is what we have done in section 2 (State of the art). To avoid confusion and to help clarify our paper, we have modified section 2’s title – that now is “Literature Review”.

In this section, we provide over 10 studies on the topic at hand (first PPPs in the wider sense, then we start narrowing the scope of the studies).

We finish in lines 70-72 with a sentence like the one proposed by the reviewer: “There isn’t however to the best of our knowledge…”.

  1. In conclusion, I propose

-evaluate the critical research, show its limitations and weaknesses,

- highlight the new knowledge and the lessons learned from it,

- describe the importance of the research and how it affects the wider field, show how the information obtained can be further used

Thanks for the critical review, we think we have addressed all the points raised.

Round 2

Reviewer 1 Report

The authors really did a great job in reworking the manuscript to be original in the sense a new research about public private partnerships can be original. English language was significantly improved. The clarity of the major definitions is much better now. The authors taken the reviewer's request as important ones, and were able to comply with that. Even defining ppp in a national context is a major help in evaluating the social scientific vlue of the article. The research is original, the ideas are fresh.  But native language proofreading could improve the situation even further.

Author Response

Thank you for re-reading our work again and for checking the changes we introduced in our paper.

In the previous round, we carried out an extensive review of our English. The paper could indeed benefit from a review done by a native English speaker. Unfortunately, we won’t have time to do this (we only have two days to re-submit our paper). We have carried out a final verification of our paper and made some minor final changes to our English – but we cannot take this any further.